# An Energy-Efficient Multi-Level Sleep Strategy for Periodic Uplink Transmission in Industrial Private 5G Networks

**DOI:** 10.3390/s23229070

**Published:** 2023-11-09

**Authors:** Taehwa Kim, Seungjin Lee, Hyungwoo Choi, Hong-Shik Park, Junkyun Choi

**Affiliations:** 1School of Information and Communication Engineering, Korea Advanced Institute of Science and Technology (KAIST), Daejeon 34141, Republic of Korea; thkim0@kaist.ac.kr; 2Institute for IT Convergence, Korea Advanced Institute of Science and Technology (KAIST), Daejeon 34141, Republic of Korea; 3School of Electrical Engineering, Korea Advanced Institute of Science and Technology (KAIST), Daejeon 34141, Republic of Korea; park1507@kaist.ac.kr (H.-S.P.); jkchoi59@kaist.edu (J.C.)

**Keywords:** private 5G, energy efficiency, small-cell base station, internet of things, reinforcement learning

## Abstract

This paper proposes an energy-efficient multi-level sleep mode control for periodic transmission (MSC-PUT) in private fifth-generation (5G) networks. In general, private 5G networks meet IIoT requirements but face rising energy consumption due to dense base station (BS) deployment, particularly impacting operating expenses (OPEX). An approach of BS sleep mode has been studied to reduce energy consumption, but there has been insufficient consideration for the periodic uplink transmission of industrial Internet of Things (IIoT) devices. Additionally, 5G New Reno’s synchronization signal interval limits the effectiveness of the deepest sleep mode in reducing BS energy consumption. By addressing this issue, the aim of this paper is to propose an energy-efficient multi-level sleep mode control for periodic uplink transmission to improve the energy efficiency of BSs. In advance, we develop an energy-efficient model that considers the trade-off between throughput impairment caused by increased latency and energy saving by sleep mode operation for IIoT’s periodic uplink transmission. Then, we propose an approach based on proximal policy optimization (PPO) to determine the deep sleep mode of BSs, considering throughput impairment and energy efficiency. Our simulation results verify the proposed MSC-PUT algorithm’s effectiveness in terms of throughput, energy saving, and energy efficiency. Specifically, we verify that our proposed MSC-PUT enhances energy efficiency by nearly 27.5% when compared to conventional multi-level sleep operation and consumes less energy at 75.21% of the energy consumed by the conventional method while incurring a throughput impairment of nearly 4.2%. Numerical results show that the proposed algorithm can significantly reduce the energy consumption of BSs accounting for periodic uplink transmission of IIoT devices.

## 1. Introduction

The number of connected Internet of Things (IoT) devices in 2023 is expected to be 16.7 billion, and it is predicted that the world will have over 30 billion connected IoT devices by 2025 [1]. This connectivity enables data collection and exchange for better insight, automation, and decision making in industries and daily life. Recently, many IoT devices require outstanding performance criteria, such as massive connectivity, throughput, ultra-low latency, and security. To fulfill these requirements, the fifth-generation (5G) provides essential functionalities such as ultra-reliable and low-latency communication (URLLC), enhanced mobile broadband (eMBB), and massive machine-type communication (mMTC) [2].

The usage of IoT devices in the industrial world is increasing rapidly, and the integration of 5G technology is expected to drive further growth in this market. According to the research report published by Future Market Insights, in 2023, it is expected that the global 5G industrial IoT market will achieve a valuation of US $1.421 million and is anticipated to reach a valuation of US $17.094 million in 2033 [3]. As industries continue to embrace the proliferation of IoT devices in the 5G network, a private 5G network is introduced to evolve industrial processes, enhance automation, improve efficiency, and pave the way for innovative industrial applications.

Private 5G networks, referred to as non-public networks by 3GPP, are physical or virtual 5G cellular systems deployed for private use to provide dedicated wireless connectivity and are isolated from public networks. Thus, private 5G networks can reduce interference from other networks and provide dedicated and localized connectivity. It allows for customization, high performance, enhanced security, and support for massive IoT connections, making them suitable for industries such as manufacturing, transportation, healthcare, and smart cities. Furthermore, the report [4] forecasts the global private 5G Network market was valued at $1348 million in 2021 and is estimated to grow at a compound annual growth rate (CAGR) of 38.8% between 2022 and 2029. This forecast aligns with the increasing demand for private 5G networks across various industries as businesses recognize the advantages of private 5G networks.

For the private 5G, overlapping the deployment of small cells is a practical and economical method to increase coverage and network performance. This strategy leads to the dense deployment of small-cell base stations (SBSs), such as ultra-dense networks (UDN), effectively addressing both coverage and performance needs. However, the massive SBS deployment significantly increases energy consumption, accounting for approximately 60∼80% of the total energy consumption in cellular networks, and this leads to an increase in operating expenses (OPEX) of the service provider [5]. Even when the base station (BS) is in an idle state, there is about 50∼60% of its maximum energy consumption [6]. As a result, to decide whether activating BSs in a particular scenario is practical and profitable, service providers must compare the higher OPEX increased by BS energy consumption with the improving network performance.

To address the optimization problem between energy saving and network performance, the BS sleep scheme is regarded as one of the most effective approaches because it is easy to deploy and does not call for changes to the existing network architecture. The BS sleep scheme can be categorized into two methods: The binary BS on/off method and the multi-level sleep modes method. The binary BS on/off scheme [7,8,9,10,11,12] is that unused or underutilized BSs are switched off to save energy. Totally shutting down BSs of binary schemes can significantly reduce energy consumption, but it might cause coverage holes in the network and have an impact on the quality of service (QoS) provided to the users. Moreover, the QoS requirements for 5G are very strict in terms of delay or packet loss.

To cope with QoS requirements and delicately control energy saving, the multi-level sleep scheme is introduced in [13,14,15,16,17]. The standardization work in [18] suggests that BSs expand the signaling period up to 160 ms so that deeper and longer sleep modes can be handled to exploit the SMs better. This makes it possible to accommodate the various sleep modes, allowing the BS to reduce its overall energy usage in 5G networks drastically. In [13], the multi-level sleep modes are proposed based on the hardware sleep capabilities such as sleep duration, transition times, activation and deactivation time, and power savings. However, when a multi-level sleep modes control scheme is used in UDN, choosing a suitable sleep mode for each SBS can be challenging because of the substantial computing cost of considering all potential sleep mode scenarios.

Thus, there have been studies based on reinforcement learning (RL) to find the optimal sleep mode of SBSs, taking into account energy efficiency (EE) and QoS constraints [15,16,19]. By leveraging RL, these approaches balance energy savings and QoS requirements, yielding practical guidelines for multi-level sleep mode settings in 5G networks and offering valuable analytical insights into configuring multi-level sleep modes effectively. However, they focus on reducing the energy consumption of downlink traffic. To provide energy-efficient IoT services, it is essential to take into account the uplink traffic [20]. Because many IoT devices generate significant uplink traffic, this trend is accelerating further in the industrial sector. In [21], the authors show the IoT traffic analysis of various industries and state that 92% of IoT devices generated more uplink than downlink traffic, and the uplink traffic is much larger than the downlink traffic in the industrial IoT applications for manufacturing.

Especially in private industrial networks, periodic uplink traffic is regularly transmitted from industrial IoT devices to centralized servers or cloud platforms. This uplink traffic allows IoT services to provide real-time insights into industrial operations, enables data-driven decision making, and facilitates predictive maintenance [22]. The IoT devices are generally programmed to sample in fixed time intervals specified by the network operator for monitoring the industrial environment. During the intervals between data transmission cycles of IoT devices, SBSs can transition to a sleep state to minimize energy consumption. Considering the periodic uplink transmission interval of each IoT device connected to BSs, it is necessary to study the optimal sleep mode.

Therefore, in this paper, we propose an energy-efficient multi-level sleep strategy for periodic uplink transmission (MSC-PUT) to optimize energy efficiency and network performance within the setup of industrial private 5G networks. The main contributions of this paper are as follows:We newly propose an MSC-PUT strategy in an industrial private 5G network that maximizes the energy efficiency of BSs. We decouple the BS on/off switching operation into three levels: active, light sleep mode (SM), and deep SM. Unlike traditional multi-level sleep mode schemes that keep sleep periods within the synchronization signal duration, we have opted for a longer deep sleep mode that extends beyond the synchronization signal period to address the considerable energy-related OPEX challenges. We formulate the energy efficiency model based on these sleep modes considering both an energy consumption model and a throughput model, including the latency caused by BS’s sleep mode.We utilize the proximal policy optimization (PPO) to address the problem of growing complexity posed by the increasing number of BSs and IoT devices in UDN and to facilitate practical implementation. We establish an MSC-PUT strategy incorporating a comprehensive PPO algorithm, considering wireless channel conditions, previous sleep mode decisions, and network traffic load. By aligning these factors with our optimization objective, MSC-PUT achieves a nearly optimal solution for managing BS sleep modes efficiently, significantly enhancing energy efficiency in the context of periodic uplink transmissions from industrial IoT (IIoT) devices in densely deployed industrial private 5G networks.We provide extensive simulations in an industrial private 5G environment from which we demonstrate that the proposed MSC-PUT achieves a substantial improvement in energy efficiency over the conventional sleep mode control schemes. We have verified that the proposed MSC-PUT algorithm achieved an energy efficiency improvement of approximately 27.5% or more while consuming less energy at 75.21% and maintaining throughput limitations to around 4.2% in comparison to the conventional multi-level sleep mode mechanism represented by *Light.*

## 2. Related Works

BSs (also known as cell towers or eNodeBs in LTE networks) are an integral part of mobile networks and consume a significant amount of energy. Various BS sleep algorithms and techniques have been studied and developed to reduce the energy consumption of BS. The algorithms aim to find the sleep operation of BSs while ensuring that coverage and service quality are not compromised. There are two different categories of sleep schemes: binary sleep modes (on and off) and multi-level sleep modes.

The binary sleep mode approach approximates the energy consumption under on/off states. Under-utilized BSs are turned off in binary sleep mode while maximizing energy efficiency. In [8], the authors propose a method to sequentially turn off a macro BS considering the downlink traffic load and the uplink traffic load. When deciding the active/sleep mode of BSs in UDN, the number of possible choices increases exponentially with the number of BSs. Recent studies have investigated incorporating machine learning approaches to address this issue. In [7], the authors formulate the traffic variations and propose a stochastic BS switching off using reinforcement learning. In [9], the authors propose an RL-based cell switching algorithm that turns off the small cells at any given time to minimize the energy consumption in ultra-dense deployments without compromising the QoS. In [10], the cumulative energy consumption over a long-term period is considered, while many conventional BS sleep techniques focused on reducing the instantaneous power consumption in UDN. In [11], the author proposes effectively reducing the action space size to operate a deep RL-based wireless network in UDN. In [12], the authors reduce the computational overhead to decide the sleep mode of BSs from a decentralized perspective and propose a multi-agent deep RL approach considering user association and user mobility in UDN.

However, when a BS switches to on/off mode in binary sleep mode, there is a transition delay between activation and deactivation time and throughput degradation. Several studies have divided BS’s sleep mode into stages to address this issue and described the energy consumption at each stage. In [13], the authors introduce four sleep modes based on an activation/deactivation time and a minimum sleep duration, enhancing network flexibility to meet traffic demands and enabling more efficient energy conservation. While a BS is not serving any user, the proposed sleep modes consist of deactivating the different components of the BS gradually. Thus, this multi-level sleep modes strategy optimizes energy savings by dynamically adjusting sleep modes in response to network activity. In [14], the authors propose a method to turn off randomly under-utilized BSs while guaranteeing QoS. The authors in [15,16] proposed a Q-learning algorithm to find the optimal duration for each sleep mode according to the energy consumption and delay constraints. The proposed approach has a gradual transition between sleep modes. In [19], the authors introduce a traffic adaptive algorithm based on an online reinforcement learning technique, enabling dynamic and direct transition decisions based on real-time traffic load. Directly transitioning from active or light sleep mode to the deepest sleep mode avoids unnecessary time wastage at intermediate levels and leads to more efficient power savings. In [17], the authors introduce the Lyapunov optimization problem to obtain the optimal BS switching result in a time series with the predicted user traffic based on long short-term memory (LSTM).

The binary on/off sleep method is simple but can result in performance problems when base stations are in the off mode, leading to coverage gaps and delays based on time spent in different modes during sleep. On the other hand, the multi-level sleep method divides energy consumption into stages, improving performance with better energy efficiency and reduced latency. However, energy savings are generally limited, excluding the deep sleep mode, which extends beyond the synchronization cycle between BS and IoT devices. As shown in Table 1, most of the existing BS sleep technique research mainly targets energy reduction during downlink transmissions in Wi-Fi or 5G networks. There is a need to study a new energy-efficient BS sleep control strategy for periodic uplink transmissions, as observed in industrial 5G networks with industrial IoT devices.

## 3. System Description

This section presents a multi-level sleep modes control for SBSs in industrial private 5G networks, specifically targeting uplink transmission. We assume that IoT devices periodically transmit uplink data to the connected BS, which can be either a macro base station (MBS) or an SBS. Initially, IoT devices are connected to the BS with the strongest signal-to-interference-plus-noise ratio (SINR). We consider the MBS to be constantly awake to maintain a sustainable connection with IoT devices. Conversely, SBSs can operate sleep mode (SM) in either light SM or deep SM to conserve energy (Figure 1).

In light SM, IoT devices maintain their connection to the SBS. The SBSs periodically receive uplink data from the IoT devices and have the opportunity to enter light SM when there is no incoming uplink data within a short timeframe. On the other hand, SBSs can enter deep SM when there is no incoming uplink data for an extended period. In deep SM, IoT devices terminate their connection to the SBS and initiate a handover process to connect to another SBS or the MBS. However, this results in a reconnection delay for IoT devices, which can compromise the QoS.

Determining an adaptive multi-level sleep mode control for SBSs is challenging due to the extensive computational complexity associated with considering all possible sleep mode cases. For instance, with 10 SBSs, the number of possible sleep mode cases amounts to $2^{10}$. Furthermore, if a long observation period (e.g., 100 timeslots) is taken into account for the multi-level sleep modes control, the number of possible sleep mode cases exponentially increases to $2^{1000}$. Such calculations are computationally infeasible for sleep mode control. To address this issue, we propose an approach that employs the RL method to design new multi-level sleep modes control for SBSs in uplink transmission (Table 2).

### 3.1. System Model

For the system model, we assume the set of BSs is B={0,1,⋯,M}, where 0 indicates the MBS and 1,⋯,M indicates the SBSs. We also assume the set of IoT devices is U={1,⋯,K}, where 1,⋯,K indicates IoT devices and $U_{m}$ is the subset of U consisting of IoT devices served by BS *m*. IoT devices transmit uplink data periodically to the connected BS with an average arrival rate of $\lambda_{k}^{t}$ during time slot *t*. The total uplink arrival rate during time slot *t* for each BS is given by,
(1)$$\Lambda_{m}^{t}=\sum_{k\in U_{m}}\lambda_{k}^{t},$$
where $\Lambda_{m}^{t}$ follows a Poisson process. According to the 5G standardization [18], the synchronization signal (SS) between IoT devices and SBSs has a period $t_{syn}\in \left\{5,10,20,40,80,160\right\}$ ms. This means that SBSs can have a light sleep time $\Delta t_{m}$ within the synchronization signal period $t_{syn}$ as defined below,
(2)$$\Delta t_{m}=min\left\{\frac{1}{\Lambda_{m}^{t}},t_{syn}\right\}.$$

As seen in Table 3, due to SM4’s extended sleep duration and the resulting disconnection between BS and IoT devices, previous multi-level sleep mode control studies cannot broadly incorporate SM4. Due to SM4’s long sleep duration and disconnection between BS and IoT devices, the previous multi-level sleep mode control studies can not widely adopt the deepest sleep mode, SM4. Since more components will be deactivated when moving BSs to a deeper SM, more energy savings can be achievable. Thus, in this paper, we utilize SM4 to maximize the energy-saving effect, and SM4 is regarded as a deep SM. SMs 1, 2, and 3 will be regarded as light SMs. A light SM can operate while maintaining a connection with a BS because it can operate within the SS interval. However, because the deep SM’s minimum sleep duration is typically longer than the SS interval, it is assumed that the connected devices must be handed over to the nearby BS when the BS is switched to deep SM. The MBS further assumes no sleep. Additionally, a periodic uplink transmission is assumed, and the IoT device’s transmission interval may alter but remains constant during the time slot.

#### 3.1.1. Average Latency

When an SBS enters sleep mode, IoT devices have a latency in processing uplink data transmission. In this subsection, we will discuss the average latency that IoT devices may experience. The average latency that IoT devices in multi-level sleep modes systems may encounter can be composed of three components: the average latency that can occur when the SBS is in a light SM state, the average latency due to the handover process caused by a deep SM state, and the average latency caused by data processing for uplink data transmission. The average latency, $D_{km}$, of the IoT device *k* that is connected to the SBS *m* in uplink data transmission is given by,
(3)$$D_{km}^{t}=\alpha_{m}^{t}\;\cdot \;\left(1-p_{m}\left(A\right)\right)\;\cdot \;D_{km}^{t,LS}+\left(1-\alpha_{m}^{t}\right)\;\cdot \;D_{km}^{t,HO}+D_{km}^{t,TX},$$
where $D_{km}^{t,LS}$ is the light SM latency of the IoT device *k* that is connected to the *m* which is in the light SM state at time slot *t*. $D_{km}^{t,HO}$ is the handover latency of the IoT device *k* due to the BS *m* being in the deep SM state at time slot *t*. $D_{km}^{t,TX}$ is the transmission latency of the IoT device *k* that is connected to the BS *m* at time slot *t*. The variable $p_{m}\left(A\right)$ is the active probability of the SBS *m*. In addition, $\alpha_{m}^{t}$ is the operation status of the BS *m* at time slot *t* which has a value 1 when the BS *m* is in the active state or the light SM state and 0 when the BS *m* is in the deep SM state.

Here, using Shannon’s information capacity theorem, the transmission latency, $D_{km}^{TX}$, of the IoT device *k* that is connected to the BS *m* is given by,
(4)$$D_{km}^{t,TX}=\frac{N_{k}^{data}}{\frac{BW}{|U_{m}|}\;\cdot \;log_{2}\left(1+\gamma_{km}^{t}\right)},\;\mathrm{where}\;N_{k}^{data}=N_{k}^{sense}\;\cdot \;c_{k},$$
where $N_{k}^{data}$ is the amount of uplink transmission from IoT device *k*, which is the product of the sensed data per second, $N_{k}^{sense}$, and the transmission periodicity, $c_{k}$. BW is the uplink channel bandwidth, and $|U_{m}|$ represents the number of IoT devices that are served by the BS *m*. Moreover, $\gamma_{km}$ is the SINR between IoT device *k* and the serving BS *m*.

The light SM latency is caused by the light SM status of BSs. The light SM period of BSs is composed of a deactivation period, a light sleep period, and a reactivation period. Figure 2 presents a cumulative distribution function (CDF) of the inter-arrival times. On the x-axis, the sequence of the BS states visited during a light SM period is reported. In particular, the labels $t_{de}$, $t_{ls}$, and $t_{re}$ on the x-axis identify the ending time of deactivation from the active state, the ending time of the light SM state, and the ending time of the reactivation state, respectively. In addition, the labels $\Delta t_{de}$, $\Delta t_{ls}$, and $\Delta t_{re}$ represent the duration of each state in the light SM period, respectively.

Hence, the probability of arrivals occurring in each state can be derived as:(5)$$\begin{array}{cc} Pr(t_{arr}\leq t_{de}) & =1-e^{-\Lambda_{m}^{t}t_{de}}\\ Pr(t_{de}<t_{arr}\leq t_{ls}) & =e^{-\Lambda_{m}^{t}t_{de}}-e^{-\Lambda_{m}^{t}t_{ls}}\\ Pr(t_{ls}<t_{arr}\leq t_{re}) & =e^{-\Lambda_{m}^{t}t_{ls}}-e^{-\Lambda_{m}^{t}t_{re}}, \end{array}$$
where $t_{arr}$ is the arrival time, and the uplink data can arrive in a deactivation state, a light SM state, and a reactivation state. The label $\Lambda_{m}^{t}$ is the total arrival rate of uplink for each BS.

The latency of IoT devices, denoted by $d_{de}$, $d_{ls}$, and $d_{re}$, for arrived data at a light SM state (deactivation state, light sleep state, activation state) can be derived as given by [23]:(6)$$\begin{array}{cc} d_{de} & =r_{de}+\Delta t_{ls}+\Delta t_{re},\;r_{de}\leq \Delta t_{de}\\ d_{ls} & =r_{ls}+\Delta t_{re},\;r_{ls}\leq \Delta t_{ls}\\ d_{re} & =r_{re},\;r_{re}\leq \Delta t_{re}, \end{array}$$
where $r_{de}$, $r_{ls}$, and $r_{re}$ represent the residual deactivation time, the residual light SM time, and the residual activation time, respectively. In our computations, we make the conservative assumption that the residual time of each state is always equal to the upper bound of the remaining time as below,
(7)$$\begin{array}{cc} r_{de} & \approx \Delta t_{de}\\ r_{ls} & \approx \Delta t_{ls}\\ r_{re} & \approx \Delta t_{re}. \end{array}$$

From Equations (Equation 5) and (Equation 6), we can derive the average light SM latency, $D^{t,LS}$, as below,
(8)$$D^{t,LS}=Pr\left(t_{arr}\leq t_{de}\right)\;\cdot \;d_{de}+Pr\left(t_{de}<t_{arr}\leq t_{ls}\right)\;\cdot \;d_{ls}+Pr\left(t_{ls}<t_{arr}\leq t_{re}\right)\;\cdot \;d_{re}.$$

Likewise, incorporating the expressions from Equation (Equation 7) in Equation (Equation 8), we can obtain the average light SM latency as:(9)$$\begin{array}{cc} D^{t,LS}= & \;\left(1-e^{-\Lambda_{m}^{t}t_{de}}\right)\;\cdot \;\left(\Delta t_{de}+\Delta t_{ls}+\Delta t_{re}\right)+\left(e^{-\Lambda_{m}^{t}t_{de}}-e^{-\Lambda_{m}^{t}t_{ls}}\right)\;\cdot \;\left(\Delta t_{ls}+\Delta t_{re}\right)\\ \; & +\left(e^{-\Lambda_{m}^{t}t_{ls}}-e^{-\Lambda_{m}^{t}t_{re}}\right)\;\cdot \;\Delta t_{re}, \end{array}$$

#### 3.1.2. Average Throughput

In this subsection, we present the throughput of IoT devices in the proposed multi-level sleep modes for periodic uplink data transmission. Throughput of uplink data from the IoT device *k* to the BS *m*, $R_{km}^{t}$, is given by,
(10)$$R_{km}^{t}=\frac{N_{k}^{data}}{c_{k}+D_{km}^{t}}=\frac{N_{k}^{sense}\;\cdot \;c_{k}}{c_{k}+D_{km}^{t}},$$
where $N_{k}^{data}$ is the transmitting data amount of the IoT device *k* in a once, and $c_{k}$ is the transmission periodicity of the IoT device *k*. The $c_{k}$ is the number of transmissions during time slot *t* since we assume that IoT *k* transmits periodically.

The $N_{k}^{data}$ is a multiplication of the sensed data per second ($N_{k}^{sense}$) and the transmission periodicity at time slot *t* ($c_{k}$). The $D_{km}^{t}$ is the average latency from Equation (Equation 3) in the previous subsection. Then, the total throughput of the BS *m* at time slot *t* is a summation of the throughput of served IoT devices as below,
(11)$$R_{m}^{t}=\sum_{k\in U_{m}}R_{km}^{t}.$$

#### 3.1.3. Energy Consumption of BS

In the proposed multi-level sleep modes for periodic uplink data transmission, the focus is on minimizing the long-term energy consumption of BSs, as opposed to conventional schemes that concentrate on minimizing instantaneous energy usage. This approach takes into account the inefficiencies that can arise from frequent mode conversions, which can lead to a significant waste of energy for the BSs. We assume a long observation time denoted as *T* to achieve this objective. The total energy consumption, $E_{total}^{T}$, of BSs during the *T* is given by,
(12)$$E_{total}^{T}=E_{MBS}^{T}+\sum_{m=1}^{M}E_{SBS,m}^{T},$$
where $E_{MBS}^{T}$, $E_{SBS,m}^{T}$ represent the energy consumption of the MBS and the SBS *m*, respectively. The energy consumption of the MBS, $E_{MBS}^{T}$, during the time *T* is given by,
(13)$$\begin{array}{cc} E_{MBS}^{T} & =\sum_{t=1}^{T}E_{MBS}^{t},\\ E_{MBS}^{t} & =(P_{MBS}^{static}+P_{MBS}^{dyn\_max}\;\cdot \;\frac{\rho_{0}^{t}}{\rho_{MBS}^{max}})\;\cdot \;\Delta t, \end{array}$$
where $E_{MBS}^{t}$ is the energy consumption of the MBS at time slot *t*. Also, $P_{MBS}^{static}$ is the static operational power of the MBS, regardless of the served IoT devices, $P_{MBS}^{dyn\_max}$ is the maximum dynamic operational power of the MBS, affected by the number of serving IoT devices. The variable $\rho_{0}^{t}$ represents the traffic load at time *t* in the MBS, while $\rho_{MBS}^{max}$ represents the maximum traffic load that the MBS can serve. On the other hand, the energy consumption of SBSs, $E_{SBS}^{T}$, during the time *T* is given by,
(14)$$\begin{array}{cc} E_{SBS}^{T} & =\sum_{t=1}^{T}E_{SBS}^{t},\\ E_{SBS}^{t} & =\alpha_{m}^{t}\left[p_{m}^{t}\left(A\right)\;\cdot \;P_{SBS}^{active}\;\cdot \;\Delta t+\left(1-p_{m}^{t}\left(A\right)\right)\;\cdot \;P_{SBS}^{ls}\;\cdot \;\Delta t\right]+\left(1-\alpha_{m}^{t}\right)\;\cdot \;P_{SBS}^{ds}\;\cdot \;\Delta t,\\ \; & =\alpha_{m}^{t}\;\cdot \;\left[p_{m}^{t}\left(A\right)\;\cdot \;\left(P_{SBS}^{static}+P_{SBS}^{dyn\_max}\;\cdot \;\frac{\rho_{m}^{t}}{\rho_{SBS}^{max}}\right)\;\cdot \;\Delta t+\left(1-p_{m}^{t}\left(A\right)\right)\;\cdot \;P_{SBS}^{ls}\;\cdot \;\Delta t\right]+\left(1-\alpha_{m}^{t}\right)\;\cdot \;P_{SBS}^{ds}\;\cdot \;\Delta t, \end{array}$$
where $E_{SBS}^{t}$ is the energy consumption of the SBS at time slot *t*. Also, $\alpha_{m}^{t}$ represents the activation status of the SBS *m* at time slot *t*, with a value of 1 when the SBS is in the active state or the light SM state, and a value of 0 when the SBS is in the deep SM state. The notations $P_{SBS}^{active}$, $P_{SBS}^{ls}$, and $P_{SBS}^{ds}$ represent the power of the SBS in the active state, the light SM state, and the deep SM state, respectively. In addition, $P_{SBS}^{static}$ is the static operational power of the SBS, regardless of the served IoT devices, and $P_{SBS}^{dyn\_max}$ is the maximum dynamic operational power of the SBS, affected by the number of serving IoT devices. The variable $\rho_{m}^{t}$ represents the traffic load at time slot *t* in the SBS, while $\rho_{SBS}^{max}$ represents the maximum traffic load that the SBS can serve. The variable $p_{m}^{t}\left(A\right)$ is the active probability of the SBS *m* at time slot *t*. Since the density of SBS is typically much higher than the density of users in UDN, we can assume that the service demands do not exceed SBS capacity. Thus, we suppose that activity during low-traffic periods is characterized as a M/M/∞ queue, and we have the following:(15)$$p_{m}^{t}\left(A\right)=1-e^{-\Lambda_{m}^{t}/\mu_{m}},$$
where $\mu_{m}$ is the average service rate of the SBS *m*.

#### 3.1.4. Energy Efficiency Maximization Problem Formulation

The energy efficiency of BSs, denoted by EE, is computed as the fraction of the throughput by energy consumption. The energy efficiency during time slot *t* is given by,
(16)$$EE^{t}=\frac{R_{total}^{t}}{E_{total}^{t}/\Delta t}=\frac{R_{MBS}^{t}+\sum_{m=1}^{m}R_{SBS,m}^{t}}{(E_{MBS}^{t}+\sum_{m=1}^{m}E_{SBS,m}^{t})/\Delta t},$$
where $R_{MBS}^{t}$ and $R_{SBS,m}^{t}$ represent the average throughput of MBS and SBS at time slot *t*, respectively, and Δt is the duration of time slot *t*. In our study, we aim to maximize the energy efficiency of BSs during time *T* in the proposed multi-level sleep modes control scheme for periodic uplink transmission. The success of an energy-efficient industrial private 5G network depends greatly on the development of a method that can efficiently govern deep SM of BSs while satisfying the rate requirements of IoT devices. Therefore, we can formally formulate the energy efficiency maximization problem as follows
(17)$$\begin{array}{cc} \max_{{\left\{\alpha^{t}\right\}}_{t=1}^{T}}\;EE & =\sum_{t=1}^{T}EE^{t}\\ s.t.\;R_{km}^{t} & \geq R_{k}^{min}. \end{array}$$

In the above-formulated problem, $\alpha^{t}$ denotes deep SM of SBSs at time slot *t* and $R_{k}^{min}$ is the minimum rate requirement of user *k* at time slot *t*. Finding the best deep SM decision of BSs that maximizes the energy efficiency of BSs is particularly challenging because the deep SM decision problem is a binary integer programming, which is one of Karp’s 21 NP-complete problems [24].

As the density of the network increases and the number of BSs grows, the complexity exponentially increases with the observed time slot. For the practical implementation, we apply an RL-based approach, which specifically uses the PPO algorithm, to find a near-optimal solution.

## 4. Multi-Level Sleep Modes Control for Periodic Uplink Transmission Strategy

In this section, we introduce a multi-level sleep modes control for periodic uplink transmission (MSC-PUT) strategy, which maximizes the energy efficiency in the industrial private 5G environment while considering latency impairment due to SM. Since the learning process in the MSC-PUT strategy is based on the PPO algorithm, we will briefly review the PPO algorithm first and then propose the MSC-PUT strategy later.

### 4.1. Basics of Proximal Policy Optimization

The PPO algorithm developed by OpenAI [25] is conceived as a state-of-the-art algorithm. It is a family of policy gradient methods for RL that alternates between sampling data by interacting with the environment and optimizing a “surrogate” objective function via stochastic gradient ascent, which makes it sample efficient [26]. It is also categorized as a model-free algorithm that does not estimate the transition probability distribution associated with the Markov decision process (MDP). Model-free RL algorithms are simpler to implement and tune than model-based RL algorithms. Therefore, we adopt the PPO algorithm for a policy learning process for the MSC-PUT strategy.

Generally, the policy π is the function that maps states to the action probability from which the agent selects an action. If the policy π is implemented by neural networks with trainable parameter θ, then we express the policy as $\pi_{\theta }$. The main goal of the policy gradient-based algorithms is learning the optimal policy, $\pi_{\theta }^{*}$, that maximizes the expected accumulated reward [27]:(18)$$\begin{array}{cc} \pi_{\theta }^{*} & =\underset{\pi_{\theta }}{\arg\max}J\left(\pi_{\theta }\right),\\ J(\pi_{\theta }) & =E_{\tau ∼\pi (\theta )}\left[\sum_{t=0}^{T}\gamma^{t}r_{t}\right]=E_{\tau ∼\pi (\theta )}\left[R(\tau )\right] \end{array}$$
where $J(\pi_{\theta })$ is the objective, τ is the trajectory from the policy, $\gamma^{t}$ is a discount factor, and $r_{t}$ is the reward. The reward for trajectory τ can be expressed as R(τ). The parameter θ is updated to maximize the objective $J(\pi_{\theta })$ as follows.
(19)$$\begin{array}{cc} \theta & \leftarrow \theta +\alpha \nabla_{\theta }J\left(\pi_{\theta }\right),\\ \nabla_{\theta }J\left(\pi_{\theta }\right) & =E_{t}\left[R_{t}\left(\tau \right)\nabla_{\theta }\log\pi_{\theta }\left(a_{t}|s_{t}\right)\right], \end{array}$$
where α is a learning rate and $\nabla_{\theta }J\left(\pi_{\theta }\right)$ is the policy gradient. In advanced algorithms, to reduce the high variance in policy gradient estimation, $R_{t}\left(\tau \right)$ in Equation (Equation 19) has been substituted to the advantage function $A_{t}^{\pi }\left(s_{t},a_{t}\right)$ which is defined as
(20)$$\begin{array}{cc} A_{t}^{\pi }\left(s_{t},a_{t}\right) & =Q^{\pi }\left(s_{t},a_{t}\right)-V^{\pi }\left(s_{t}\right),\\ \nabla_{\theta }J\left(\pi_{\theta }\right) & =E_{\tau ∼\pi_{\theta }}\left[A_{t}^{\pi }\left(s_{t},a_{t}\right)\nabla_{\theta }\log\pi_{\theta }\left(a_{t}|s_{t}\right)\right], \end{array}$$
where $Q^{\pi }\left(s_{t},a_{t}\right)$ is the action-value function and $V^{\pi }\left(s_{t}\right)$ is the state-value function which can be parameterized by the critic network [28].

The main idea of the PPO algorithm is that it improves the policy monotonically to avoid a performance collapse and makes it easier to implement with clipping. The surrogate objective function was introduced in trust region policy optimization (TRPO) [26] as
(21)$$J^{CPI}\left(\theta \right)=E_{t}\left[\frac{\pi_{\theta }\left(a_{t}|s_{t}\right)}{\pi_{\theta_{old}}\left(a_{t}|s_{t}\right)}A_{t}^{\pi_{\theta_{old}}}\right]=E_{t}\left[r_{t}\left(\theta \right)\right],$$
where CPI refers to conservative policy iteration and $r_{t}\left(\theta \right)$ is the probability ratio. This objective function is maximized within a constraint on the size of the policy change, which is expressed as
(22)$$\begin{array}{cc} \max_{\theta }\; & E_{t}\left[r_{t}\left(\theta \right)A_{t}^{\pi_{\theta_{old}}}\right],\\ s.t.\; & E_{t}\left[KL(\pi_{\theta }\left(a_{t}|s_{t}\right)||\pi_{\theta_{old}}\left(a_{t}|s_{t}\right))\right]\leq \delta , \end{array}$$
where KL(a||b) represents KL divergence between the probability *a* and *b*, which measures the difference between two probability distribution. The adaptive KL penalty PPO has been suggested as the unconstrained optimization problem with using KL divergence as a penalty rather than a constraint, which is given by
(23)$$J^{KLPEN}\left(\theta \right)=E_{t}\left[r_{t}\left(\theta \right)A_{t}^{\pi_{\theta_{old}}}-\beta KL(\pi_{\theta }\left(a_{t}|s_{t}\right)\left|\right|\pi_{\theta_{old}}\left(a_{t}|s_{t}\right))\right],$$
where β is a penalty coefficient. Since it is difficult to decide the value of β, an adaptive coefficient application method was proposed. However, its performance is inferior to that of the clipping method. The clipping method removes the KL divergence constraint and makes the objective more simple, which is represented as
(24)$$J^{CLIP}\left(\theta \right)=E_{t}\left[\min\left(r_{t}\left(\theta \right)A_{t}^{\pi_{\theta_{old}}},clip\left(r_{t}\left(\theta \right),1-\epsilon ,1+\epsilon \right)A_{t}^{\pi_{\theta_{old}}}\right)\right].$$

The clip function restricts the value of $J^{CPI}\left(\theta \right)$ between $\left(1-\epsilon \right)A_{t}^{\pi_{\theta_{old}}}$ and $\left(1+\epsilon \right)A_{t}^{\pi_{\theta_{old}}}$. The objective function $J^{CLIP}\left(\theta \right)$ prevents parameter updates that can cause the policy to alter quickly and unstably. It is a sample-efficient method since $J^{CLIP}\left(\theta \right)$ makes it possible to reuse the extracted trajectory many times. Furthermore, it simplifies the implementation of the algorithm.

### 4.2. Multi-Level Sleep Modes Control for Periodic Uplink Transmission Model

The MSC-PUT strategy, which is controlled by an agent on the MBS side, centrally manages SBSs. SBSs work probabilistically in active and light SM while keeping a connection with BS, according to the model given forth in the previous section. Using these models as a basis, the MSC-PUT strategy learns policies that determine deep SM and maximize the network’s energy efficiency in a given situation. Figure 3 depicts the overall operational process of the MSC-PUT strategy.

The goal of the MSC-PUT strategy is to find the deep SM that can maximize energy efficiency by balancing the throughput of MBS and SBS with respect to their energy consumption. Latency due to sleep mode is defined as a performance degradation factor in Equation (Equation 3) in order to create an energy efficiency model based on average throughput and energy consumption.

SBS maintains an active state but enters a light SM when it is in an idle state without transmitting data. Furthermore, the agent in MBS makes decisions about SBSs’ deep SM by using reinforcement learning based on the PPO algorithm in order to maximize energy efficiency and applies it to the network. Note that MBS is constantly active in order to guarantee IoT service in the MSC-PUT strategy.

This subsection defines the state, action, and reward function for the MSC-PUT strategy.

***State***: State is the information that an agent acquires through observing its environment. We define the state of the environment as
(25)$$\begin{array}{c} s^{t}≜\left[s_{0}^{t}\;\;\cdots \;\;s_{m}^{t}\;\;\cdots \;\;s_{M}^{t}\right],\\ \mathrm{where}\;s_{m}^{t}=\left[{\left(\Gamma_{m}^{t-1}\right)}^{T}\;\;{\left(\Gamma_{m}^{t}\right)}^{T}\;\;\alpha_{m}^{t-1}\;\;\rho_{m}^{t}\right], \end{array}$$
which is the concatenation of the individual BS’s state, $s_{m}^{t}$. It should consist of major features that represent the current state of the environment well to train the good policy. Firstly, to take account of the temporal correlation of the wireless channel, the state contains SINR between IoT device *i* and BS *j* at time slot t−1 and *t* as $\Gamma^{t-1}$ and $\Gamma^{t}$, where $\Gamma^{t}$ is K×1 matrix of $\gamma_{k,m}$ as
(26)$$\Gamma^{t}=\left[\begin{array}{c} \gamma_{1,m}^{t}\\ \vdots \\ \gamma_{K,m}^{t} \end{array}\right].$$In addition, the decision of deep SM at time slot t−1 is included in states to represent the previous SMs of all SBSs, $\alpha_{m}^{t-1}$. Lastly, the traffic load of all BSs is also an essential feature in deciding deep SM. Therefore, the traffic load vector at time slot *t* is included as a state element, $\rho_{m}^{t}$.***Action***:
(27)$$a^{t}≜\alpha^{t}=\left[\alpha_{1}^{t}\cdots \alpha_{M}^{t}\right],$$The action is the decision of whether SBSs switch to deep SM or not. Thus, the action is defined as above where $\alpha_{m}^{t}$ is the indicator. If SBS *m* is in active mode or light SM at time slot *t*, then $\alpha_{m}^{t}$ is 1, or if SBS *m* is in deep SM, then it is 0. The action space is the combination of SBSs’ deep SM, and it becomes $2^{M}$.***Reward***:
(28)$$r^{t}≜EE^{t}-\varepsilon^{t},$$The reward function is defined as above. It consists of energy efficiency and penalty terms. The learning agent attains energy efficiency based on its decision during time slot *t*. Furthermore, to adhere to the constraint presented in Equation (Equation 17), the reward function incorporates a penalty term, $\varepsilon^{t}=q\sum_{k\in U}1_{\left\{R_{km}^{t}<R_{k}^{min}\right\}}$, where *q* is the regularization coefficient for the penalty. The penalty is proportional to the count of IoT devices unable to meet the minimum rate requirement at time slot *t*. Therefore, the reward maximization problem is equally valid as the problem of Equation (Equation 17).

The deep learning parameters must be optimized throughout training to ensure that the MSC-PUT strategy performs properly and maximizes energy efficiency. The MSC-PUT strategy’s detailed training process is depicted as a pseudo-code in Algorithm 1. Firstly, the parameter of the actor network and critic network is initialized for a given private 5G environment setting. From lines 4 to 11, using the prior actor network, the agent gathers trajectory data and computes target value and advantage. The agent observes states of the environment, which are defined in Equation (Equation 25), and determines with policy $\theta_{A_{old}}$ which SBS to switch to deep SM. Subsequently, each SBS computes its uplink throughput and energy consumption. The agent then obtains the uplink throughput and energy consumption of each SBS and computes the reward $r_{t}$. After that, the trajectory is kept in the batch containing advantage $A_{t}$ and target value $V_{tar,t}^{\pi }$ in order to extract it as a sample later. From lines 12 to 22, based on the PPO algorithm, the parameters of the actor network and critic network are updated during the loop for epoch *L*.
**Algorithm 1** Training process of MSC-PUT1:**Given:**   B={0,1,⋯,M}: set of BSs which *m* SBSs over one MBS   U={1,2,⋯,K}: set of IoT devices,   IoT devices transmit uplink data periodically with an average arrival rate $\lambda_{k}^{t}$   $U_{m}$: subset U and IoT devices which are served by BS *m*   The BS $B_{m}$ receives uplink data $\Lambda_{m}^{t}$ from connected IoT devices $\Lambda_{m}^{t}=\sum_{k\in U_{m}}\lambda_{k}^{t}$   SBSs have average light sleep time $\Delta t_{m}$ within a synchronization signal period $t_{syn}$    $\Delta t_{m}=min\left\{\frac{1}{\Lambda_{m}^{t}},t_{syn}\right\}$2:**Initialization:**   Initialize $\theta_{A}$, $\theta_{C}$   Set learning rates of actor and critic $\alpha_{A}$, $\alpha_{C}$   Set the number of episode *N* and epoch *L*3:**for** Episode = 1,2,⋯,N **do**4:   **for** t=1:T **do**5:     set $\theta_{A_{old}}=\theta_{A}$6:     $s^{t}=\left[s_{0}^{t}\;\;s_{1}^{t}\;\;\cdots \;\;s_{M}^{t}\right]$7:     $a^{t}=$ Select action with policy $\theta_{A_{old}}$ in state $s^{t}$8:     Apply action $a^{t}$ and get $r^{t}$9:     Calculate $A^{t}$, $V_{tar,t}^{\pi }$10:     Store $(s_{t},a_{t},r_{t},A_{t}$, $V_{tar,t}^{\pi })$ in the batch11:   **end for**12:   **for** Epoch = 1,2,⋯,L **do**13:     **for** mini batch *m* in the batch **do**14:        Calculate $r_{m}\left(\theta_{A}\right)$, $A_{m}$, $J_{m}^{CLIP}\left(\theta_{A}\right)$15:        Calculate entropy $H_{m}$ using $\theta_{A}$16:        Calculate policy loss: $L_{pol}\left(\theta_{A}\right)=J_{m}^{CLIP}\left(\theta_{A}\right)-\beta H_{m}$17:        $\theta_{A}\leftarrow \theta_{A}+\alpha_{A}\nabla_{\theta_{A}}L_{pol}\left(\theta_{A}\right)$18:        Calculate predicted ${\hat{V}}^{\pi }\left(s_{m}\right)$ using $\theta_{C}$19:        Calculate value loss: $L_{val}\left(\theta_{C}\right)=MSE\left({\hat{V}}^{\pi }\left(s_{m}\right),V_{tar,t}^{\pi }\left(s_{m}\right)\right)$20:        $\theta_{C}\leftarrow \theta_{C}+\alpha_{C}\nabla_{\theta_{C}}L_{val}\left(\theta_{C}\right)$21:     **end for**22:   **end for**23:**end for**24:**Output:**$\theta_{A},\theta_{C}$

### 4.3. Complexity Analysis

In this subsection, we analyze the time complexity of the MSC-PUT strategy. In order to quantify the complexity of the MSC-PUT strategy, we compute the time complexity of the actor networks. If we feed forward from layer *i* to *j*, $S_{j}=W_{ji}*Z_{i}$, then $S_{jk}=W_{ji}*Z_{ik}$ of which operation has O(j∗i∗k) time complexity. In addition, if we apply activation function $Zjk=f(S_{jk})$, then it has O(j∗k) time complexity, because of the element-wise operation. So, the total time complexity becomes O(j∗i∗k+j∗k). In the case of backward propagation, the time complexity is the same as that of the feed-forward O(k∗j+k∗i∗j) As a result, when we serialize the vector, it becomes O(2jk) for training or inference.

For the actor network of the MSC-PUT strategy, the dimension of the input vector becomes $R^{\left(2MK+2(M+K)+1\right)}$ by serialization of the state vector in Equation (Equation 25), and the dimension of output vector becomes $R^{\left(2^{M}-1\right)}$, as in Equation (Equation 27). With *N* hidden layers and ω nodes per hidden layer, the actor network’s time complexity corresponds to $O(2\omega \left(\left(2MK+2\left(M+K\right)+1\right)+2N\omega +2^{M}\right)$.

## 5. Performance Evaluation

### 5.1. Simulation Setup

In this section, we present the evaluation of the performance of the proposed MSC-PUT algorithm using the Python and the PyTorch library for the PPO algorithm implementation [26]. Our simulation considers an indoor factory IoT scenario where *M* SBSs are deployed over one MBS and serve moving *K* IoT devices. The total 18 SBSs are located on the square lattice with spacing 50 m between SBSs and 25 m from the walls in the 300 m by 150 m sized hall as the indoor factory scenario in [29]. The MBS is located at the center of the hall, and it is always in active mode. The simulation layout is presented in Figure 4. The text in Figure 4 denotes the pairs of IoT devices’ ID and BS’s ID, which represent the connection between the IoT device and the BS. The IoT devices move freely at a constant speed of 1 m/s in the hall and send collected data through the uplink. We assumed that IoT devices collect 1 Mbit of data per second. Thus, IoT device *k* sends the size of 1 Mbit/$\lambda_{k}^{t}$ burst data to the connected BS, where the arrivals of the burst data follow the Poisson process and $\lambda_{k}^{t}$ is the average number of arrivals.

For the path loss model, we followed the indoor factory with sparse clutter and high BS height (InF-SH) scenario among the indoor factory scenarios which are introduced in [29]. Thus, we used the NLOS path loss model where the path loss between IoT device *k* and BS *m*, $PL_{NLOS}^{k,m}$ is given by
(29)$$\begin{array}{cc} PL_{NLOS}^{k,m} & =max(PL^{k,m},PL_{LOS}^{k,m}),\\ PL^{k,m} & =32.4+23.0\log_{10}\left(d_{3D}\right)+20\log_{10}\left(f_{c}\right),\;\sigma_{SF}=5.9,\\ PL_{LOS}^{k,m} & =31.84+21.5\log_{10}\left(d_{3D}^{k,m}\right)+19\log_{10}\left(f_{c}\right),\;\sigma_{SF}=4.0, \end{array}$$
where the three-dimensional distance $d_{3D}^{k,m}=\sqrt{{\left(d_{2D}^{k,m}\right)}^{2}+{\left(h_{m}-h_{k}\right)}^{2}}$, $1\leq d_{3D}\leq 600$ m, $f_{c}$ is the center frequency normalized by 1 GHz and $\sigma_{SF}$ is the standard deviation of shadow fading.

The MCS-PUT strategy decides whether SBSs switch into deep SM or not per every time slot. The time slot consists of multiple sub-time slots with the same duration as the SS interval in 5G NR, which can be set to {5,10,20,40,80,160} ms. We set the duration of a time slot to 1s for {5,10,20,40} ms SS intervals; thus, there are {200,100,50,25} sub-slots in a time slot, respectively. On the other hand, the time slot is set to 1.04 and 1.12s for {80,160} ms to match the cycle and {13,7} sub-slots are in a time slot, respectively. We construct two independent, fully connected multi-layer perceptron (MLP) networks for the actor and critic networks with two hidden layers of 256 nodes for each layer to implement the MCS-PUT strategy. We employ the ReLU activation function and use the ADAM optimizer. The learning rate for both the actor and critic networks is set to $10^{-5}$, and the discount factor is configured at 0.99. Additionally, the batch size is set to 64. The MLP networks are trained for 500 episodes of the same scenario and evaluated after the training.

The scenario consists of 200 time slots, which means the actual observation time duration is 60 s for {5,10,20,40} ms SS intervals, 208 s for 80 ms SS intervals, and 232 s for 160 ms SS intervals.

The detailed values of parameters used in the simulation are summarized in Table 4.

The performance was verified by changing the values of the number of IoT devices in the network, the arrival parameter ($\lambda_{k}$), and the SS interval. We obtained the average of the 60 time slots for each experiment and compared the averages of multiple experiments with various random seeds.

### 5.2. Benchmarks

The performance of our algorithm is compared with four different BS sleep operations, which can produce some reasonable results.

***AlwaysOn***: Always-on scenario. All BSs are active and never switch into sleep mode.***Light***: Light sleep only scenario. BSs only switch into light SM, which serves as a benchmark for conventional multi-level sleep mode methods referred from [13,14], where BSs progress through sleep mode stages sequentially until they transition to an awake mode. Deep SM is not employed in this process.***NoConn***: Deep sleep when there is no connection scenario. BSs can switch into light SM and switch into deep SM only when there are no connected IoT devices.***Threshold***: Deep sleep with threshold scenario. BSs can switch into light SM and switch into deep SM only when a BS serves fewer IoT devices than the average number of IoT devices per BS.

Furthermore, we conducted a benchmark comparison *Binary* with the conventional binary on/off method referred from [7,10], applying deep SM as the off mode without engaging in light SM. We then compared this approach with our proposed method in Table 5. We perform the comparison against the number of IoT devices, average number of arrivals, and sub-slot time, and all these methodologies are implemented using the Python language.

### 5.3. Simulation Results

#### 5.3.1. Convergence of the Algorithm

We present the log probability, critic value, and reward graphs during the training process of a sample scenario in Figure 5 to demonstrate the proposed algorithm’s convergence. We train the model for 500 episodes, which consist of 200 time slots; thus, there are $10^{5}$ train steps. The values on the graph are computed using a moving average every 50 steps. Since the algorithm deploys both neural networks, we investigate the learning curves of the log probability for the actor network and the critic values for the critic network. To make sure the entire system works properly, we also examine the learning curve of the rewards during the training period.

As part of the training of the parameter of the actor network ($\theta_{A}$), Figure 5a displays the log probability of the chosen action in a certain state of a particular trajectory. The fact that the log probability converges to 0 shows that $\theta_{A}$ has undergone sufficient training since it means that the actor network chooses the action that has the highest probability of being perceived as optimal in a given state. Figure 5b shows that the value of a given state becomes saturated as the critic network’s parameter ($\theta_{C}$) converges. Furthermore, the reward achieves convergence at about $3.2\times 10^{4}$ steps in Figure 5c. These results imply that the proposed algorithm’s learning process is working properly.

#### 5.3.2. Performance by the Number of IoT Devices

In Figure 6, Figure 7 and Figure 8, we discuss and evaluate the performance of the proposed algorithm by varying the number of IoT devices in scenarios where the arrival rate is set to 50 and slot time is set to 0.16 ms to examine the impact of the number of IoT devices on performance. The UDN is a network where the density of BS is significantly higher than the density of users, and the highest number of active users is around 600 when there are $10^{3}$ BSs per 1 km${}^{2}$, according to the concept of UDN given in [30,31]. So, we set the range of the active IoT device population between 10 and 60 in the design of our simulation shown in Figure 4.

Note that the additional latency brought on by sleep mode is represented by throughput impairment because the throughput defined here is the transfer rate against the additional delay brought on by SM. The likelihood of being in light SM diminishes with the number of IoT devices. Hence, the gap in throughput performance between *Light* and *AlwaysOn* scenarios reduce as more IoT devices are given to the network. For the same reason, Figure 7 shows that scenario *Light*’s energy consumption increases slightly as the number of devices increases. The throughput performance of *NoConn* is the same as *Light*. Since only SBSs without served IoT devices switch into deep SM, there are no additional factors for inducing handover and transmission latency. In contrast, when the number of devices increases, the throughput for *Threshold* and *MSC-PUT* tends to decline. To maximize the energy efficiency, *Threshold* and *MSC-PUT* switch to deep SM and hand over the connected devices to nearby BSs even when servicing IoT devices are present. The average number of IoT devices connected per BS increases as the network’s number of devices rises. As a result, when BSs switch into deep SM, a considerable amount of IoT devices undertake handover in *Threshold* and *MSC-PUT* scenarios. And throughput drops as the number of devices increases because the impact of handover latency grows. Furthermore, handed-over devices experience additional latency since they must transmit to a BS located further away than usual. However, compared to *Light*, the scenario’s throughput reduction in deep SM (*Threshold* and *MSC-PUT*) is only about maximum 4.1%, while the decrease in energy consumption is at least 11.9%, making the energy reduction superior to the throughput reduction. Thus, energy efficiency is improved. When the number of connected devices increases, the possibility that an IoT device is not connected to the BSs decreases. Thus, although *NoConn* also uses deep SM, *NoConn*’s energy efficiency performance converges to *Light*’s as increasing energy consumption. Figure 8 shows that *MSC-PUT* maximizes energy efficiency performance from 43.4% to 36.2% than *Light* by reducing energy consumption by up to 18.7% while maintaining throughput impairment within 4.1%.

#### 5.3.3. Performance by Arrival Rates

In Figure 9, Figure 10 and Figure 11, we analyze the effect of the arrival rates on performance by examining and assessing the performance of the proposed method in scenarios where the number of IoT devices set to 50 and slot time is set to 0.16 ms. The average light sleep time increases when the arrival rate is low since the average light sleep time of BS *m* equals $\frac{1}{\Lambda_{m}^{t}}$. Also, the lower arrival rates cause an increase in the average latency time for arrivals during the light sleep period. As a result, throughput generally inclines as the arrival rate increases. *Light* and *NoConn* scenarios show the same throughput tendency that approaches the throughput of *AlwaysOn* as arrival rates increase. Throughput of *Threshold* shows a similar trend of *Light* and *NoConn*, but it is degraded more than that of the two scenarios because *Threshold* scenario triggers handover, which causes handover latency and also might cause more transmission latency. The throughput performance of *MSC-PUT* also tends to improve as the arrival rate grows since the chance of being in light SM decreases. However, to balance the performance of reducing energy consumption against the degradation of throughput performance, the *MSC-PUT* switches more SBSs into deep SM. This leads to more handovers than in the other benchmark settings, which worsens throughput performance as in Figure 9.

Arrival rates in the simulation do not affect the total energy consumption in benchmark scenarios because the sum of energy consumption of BSs mostly depends on the number of IoT devices, and we assume that IoT devices send the same amount of traffic with the same arrival rates for every second to BSs in this simulation. As can be seen in Figure 10, light SM can reduce 24.6%. The amount that BSs go into deep SM primarily impacts how much energy can be saved further. The improvement in energy consumption of *NoConn* is modest compared to *Light* because there are many IoT devices in a given experimental setup and few BS without devices. In contrast, *Threshold* more aggressively switches to deep SM than *NoConn*, and it reduces energy consumption by about 10%. Moreover, *MSC-PUT* reduces energy consumption further than *Threshold*, about 13.5%, at the expense of introducing latency, which impairs throughput performance.

In conclusion, as it influences the length of light sleep, arrival probability, and latency time at the moment of arrival, the arrival rate is a metric intimately related to throughput performance. Throughput impairment diminishes as the arrival rate rises, and energy efficiency even somewhat improves as the arrival rate rises. In addition, using deep SM, *MSC-PUT* shows an energy efficiency improvement of up to 35.84% compared to *Light*.

#### 5.3.4. Performance by Sub-Slot Time

We examine and evaluate the performance of the proposed method in scenarios where the number of IoT devices is set to 50, and the arrival rate is set to 50 to figure out the impact of the sub-slot time on performance in Figure 12, Figure 13 and Figure 14. Note that the sub-slot period is identical to the SS interval in 5G NR, as we mentioned in the previous subsection regarding the simulation configuration. As we can see from Figure 12, there is a clear correlation between slot time and throughput. As slot time increases, throughput decreases. This is because light SM, which operates while keeping a connection with the BS, works within the slot time. As a result, the slot time limits the maximum light sleep time. The average wake-up delay time increases when the maximum light sleep time and slot time increase. This can lead to a decrease in throughput performance in the given arrival rate conditions.

Due to the effect of longer light sleep duration in Figure 12, the relative throughput performance of *Light* and *NoConn* degrades as sub-slot time increases. On the other hand, the relative throughput performance of *Threshold* and *MSC-PUT* tends to improve as the slot time grows. This happens because a handover results from BSs’ deep sleep, which causes an aggregated arrival rate and reduces average light sleep time. The energy consumption is dependent on the number of IoT devices, the same as the previous performance by arrival rates. Therefore, slot time has no impact on the performance of energy consumption.

In summary, the throughput performance declines as the average light sleep duration grows with increasing sub-slot time, and correspondingly, so does the energy efficiency performance. However, aggregating the arrival rates due to deep SM and handover can improve from 64.54% to 66.04% *MSC-PUT*’s performance compared to scenario *AlwaysOn* as the sub-slot period increases.

In Table 5, we compare the performance evaluation of MSC-PUT and other algorithms with respect to throughput, energy consumption, and energy efficiency under the following conditions: 50 IoTs, an arrival rate of 50, and sub-time slots of 0.16 ms. Furthermore, MSC-PUT’s performance improvement is confirmed by comparing it with *AlwaysOn*, *Light*, *NoConn*, *Threshold*, and *Binary*. To compare *MSC-PUT* against conventional sleep operation, *Light* conceptually represents the conventional multi-level sleep operation, and *Binary* represents the conventional binary on/off method. The numerical values below each comparison algorithm represent the percentage of performance improvement of MSC-PUT compared to the respective comparison algorithm. When considering the *Light*, latency resulting from the operation of SM1, SM2, and SM3 in the conventional multi-level sleep operation is calculated as average values using our model. As shown in Table 5, it is observed that our proposed MSC-PUT consumes energy at 75.21% of the energy consumed by the *Light* method while incurring a throughput impairment of nearly 4.2%. Because *MSC-PUT* employs the most extended deep sleep mode, it demonstrates a substantial reduction in BS energy consumption, and *MSC-PUT* enhances energy efficiency by nearly 27.5% when compared to *Light*. *NoConn* and *Threshold* also employed deep sleep mode, but *MSC-PUT* achieves a lower energy consumption, consuming only 76.52% and 89.3% compared to them, respectively. This is because *MSC-PUT* considers the maximizing of energy efficiency using the PPO algorithm, taking into account the trade-off between throughput impairment and energy savings caused by BS sleep operation. In pursuit of maximizing energy efficiency, throughput performance incurred a slight loss of up to approximately 4.8% compared to *AlwaysOn*, *Light*, *NoConn*, and *Threshold*. However, in the proposed *MSC-PUT*, the energy efficiency model formulation includes rate requirements as constraints, making it suitable for providing IIoT applications. *Binary* exhibits approximately 11% better performance in terms of energy consumption than *MSC-PUT* but also demonstrates the most significant throughput impairment, at around 25%. And *MSC-PUT* outperforms energy efficiency by 12.74% compared with *Binary*.

## 6. Discussion

Our proposed approach improves energy efficiency by applying sleep mode under conditions that meet the requirement rate. However, compared to conventional multi-level sleep mode or binary on/off, the proposed MSC-PUT introduces a slight throughput reduction. This aligns with the fact that IoT service requirements in mixed-criticality industrial environments, as described in 3GPP document [32]. For instance, motion control or automated guided vehicles demand up to 100mbps peak data rates, whereas process and asset monitoring require approximately 1Mbps. Additionally, due to the high cost of 5G devices, there is research on reduced capability (RedCap), also known as NR-Light, which caters to cases where high-performance specifications are not essential, even within a 5G network. As complexity decreases, NR-Light devices become more cost-effective, consume less power, extend battery life, and reduce device size, offering new possibilities. Considering the throughput impairment in the proposed MSC-PUT, even with rate requirement constraints, it seems better suited for applications with slightly relaxed rate requirements. This encourages further research on energy-efficient sleep mode control for IIoT applications that may require higher data rates in the future.

Similarly to other papers [7,33,34], we have also utilized the assumption of Poisson-distributed input traffic for our input traffic, potentially limiting its applicability beyond this model. In practice, we assumed the BS’s buffer size to be infinite, making the input traffic volume manageable and independent of the input traffic model. As we have constrained the optimization to meet rate requirements, we cautiously speculate it can be applied with a large buffer size in the simulation. This assumption accelerates research in constrained environments where factors like user input traffic and buffer size are limited in real-environment settings.

## 7. Conclusions

In this article, we investigated an energy-efficient MSC-PUT strategy in industrial private 5G networks. We considered the periodic uplink transmission of IIoT devices and formulated an energy efficiency model taking into account the trade-off between throughput impairment and energy efficiency due to MSC-PUT. The decision problem of BS deep sleep modes is NP-complete problems in dense deployment of BSs. To tackle the problem, we applied a PPO algorithm, an RL-based approach, to find a near-optimal solution. By conducting experiments, it was possible to evaluate and examine the performance of benchmark scenarios in terms of the number of IoT devices, arrival rates, and sub-slot time. As a result, we have confirmed that in comparison to the conventional multi-level sleep mode mechanism represented by *Light*, the proposed *MSC-PUT* algorithm achieved an energy efficiency improvement of approximately 27.5% or more while consuming less energy at 75.21% and maintaining throughput limitations to around 4.2%. Furthermore, when compared to the conventional binary on/off mechanism represented *Binary*, *Binary* outperforms energy consumption performance of approximately 11% than *MSC-PUT* but also introduces the most significant throughput impairment, at around 25%. *MSC-PUT* outperforms energy efficiency by 12.74% compared with *Binary*. Our proposed *MSC-PUT* greatly outperforms the compared method in terms of energy efficiency.

## Figures and Tables

**Figure 1 sensors-23-09070-f001:**
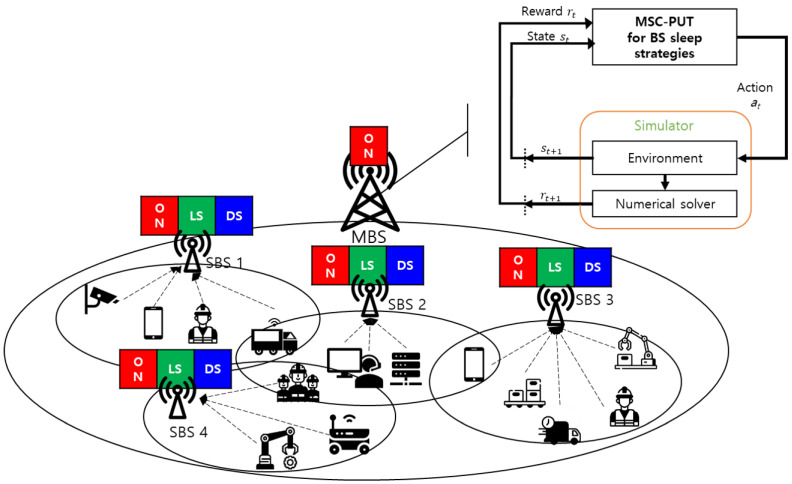
Multi-level BS sleep strategy for industrial IoT devices.

**Figure 2 sensors-23-09070-f002:**
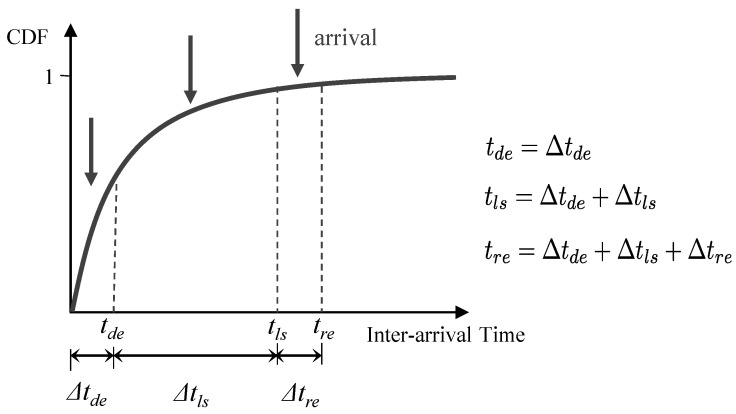
Distribution of inter-arrival times comprising light SM.

**Figure 3 sensors-23-09070-f003:**
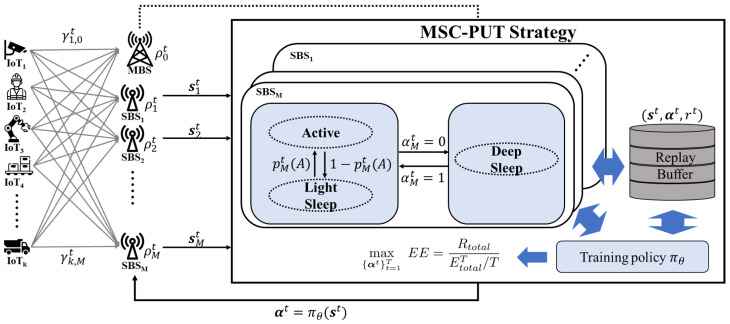
An overview of the MSC-PUT operation.

**Figure 4 sensors-23-09070-f004:**
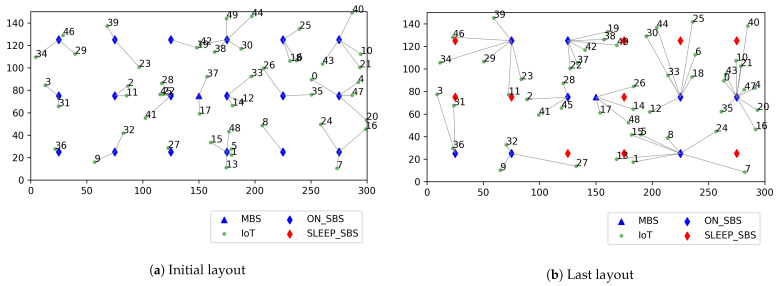
Example of simulation layout.

**Figure 5 sensors-23-09070-f005:**
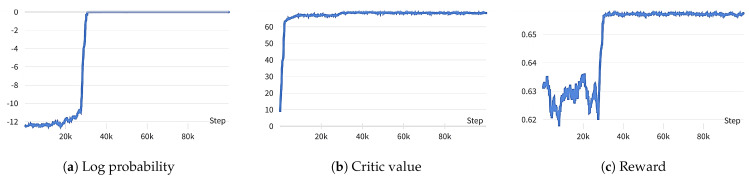
Convergence of MSC-PUT algorithm.

**Figure 6 sensors-23-09070-f006:**
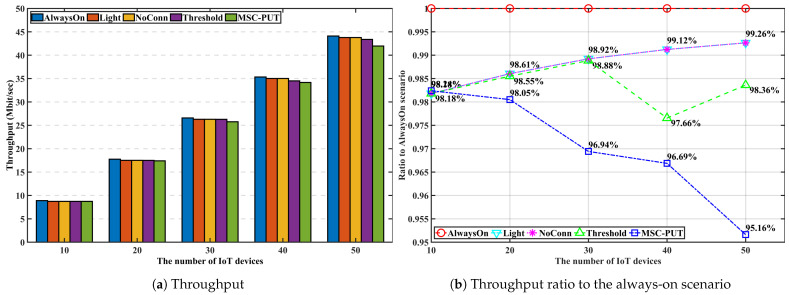
Throughput performance according to the number of IoT devices.

**Figure 7 sensors-23-09070-f007:**
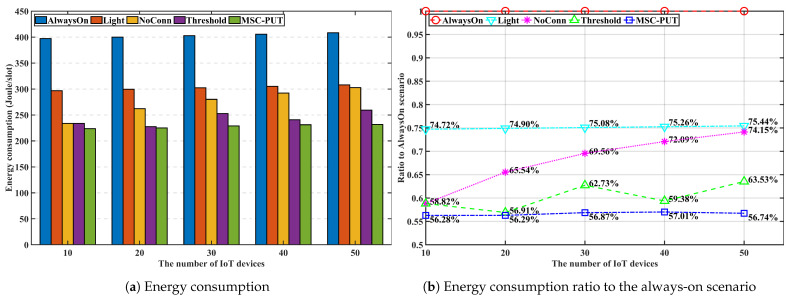
Energy consumption performance according to the number of IoT devices.

**Figure 8 sensors-23-09070-f008:**
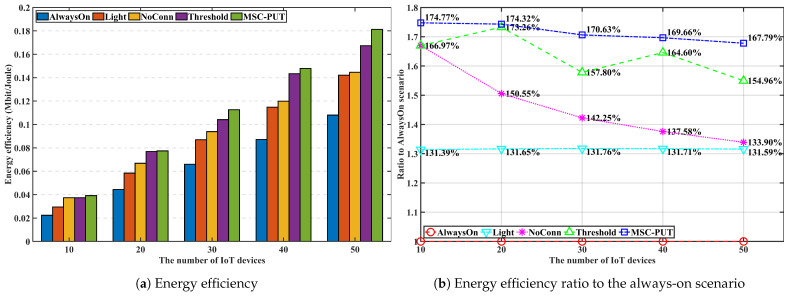
Energy efficiency performance according to the number of IoT devices.

**Figure 9 sensors-23-09070-f009:**
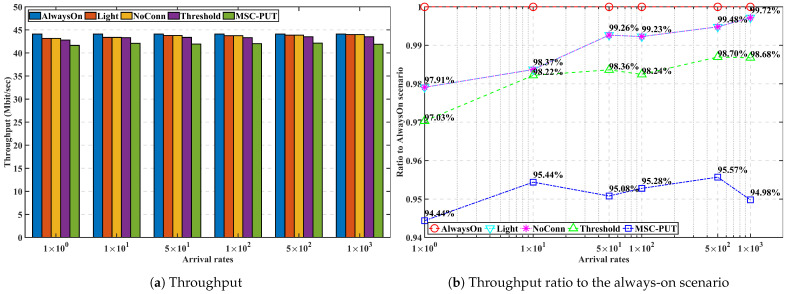
Throughput performance according to the arrival rates.

**Figure 10 sensors-23-09070-f010:**
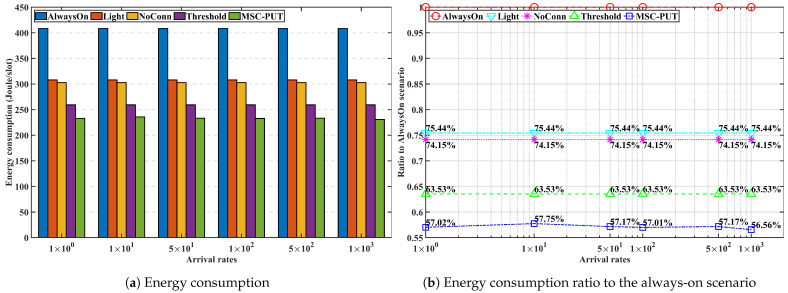
Energy consumption performance according to the arrival rates.

**Figure 11 sensors-23-09070-f011:**
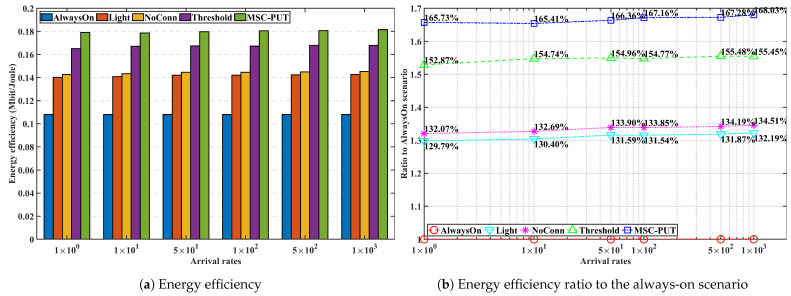
Energy efficiency performance according to the arrival rates.

**Figure 12 sensors-23-09070-f012:**
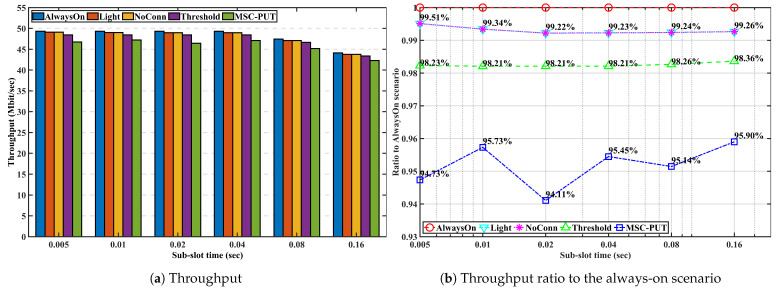
Throughput performance according to the slot time.

**Figure 13 sensors-23-09070-f013:**
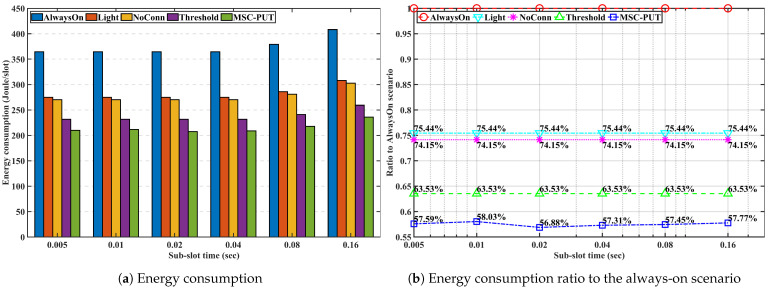
Energy consumption performance according to the slot time.

**Figure 14 sensors-23-09070-f014:**
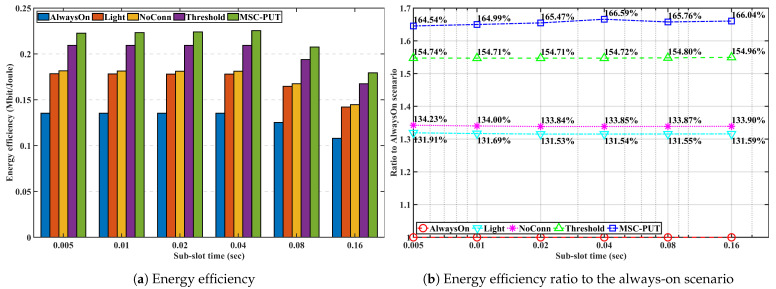
Energy efficiency performance according to the slot time.

**Table 1 sensors-23-09070-t001:** Comparison of related works.

Sleep Mode	Related Work	Description	Advantage	Limitation
Binary	[7]	Provide BS on/off operation to match up with traffic load variation using RL	Minimize energy consumption of BSs with fast ongoing learning process	Only adopt to limited scenarios Sparse deployment of BSs
	[8]	Sequentially turn off the macro BSs	Consider both downlink and uplink traffic	Macro BS
	[9]	Provide traffic-aware BS switching method using DQL (Deep Q-learning)	Energy saving without compromising QoS	Consider downlink transmission
	[10]	Provide LSTM based BS on/off decision in UDN	Reduce cumulative energy consumption	Not employ deep sleep
	[11,12]	Provide DRL-based approach to reduce energy consumption in UDN	Reduce the computational overhead to decide BS switching operation	Not employ deep sleep Consider only downlink transmission
Multi-level	[13]	Provide multi-level sleep modes	Enhance the network flexibility to meet traffic demands	Unscalable within UDN
	[14]	Try to optimal proportion of sleep BSs based on stochastic geometry	Optimize sleep mode under random sleeping with low computational complexity	One small cell
	[15,16]	Provide the optimal duration of sleep mode using Q-learning algorithm	Adjust sleep mode models according to the energy consumption and delay constraints	Consider only downlink traffic
	[17]	Provide LSTM based user traffic prediction	Reduce the synchronization overhead between BSs and users by forecasting	Sparse deployment of BSs
	[19]	Propose a traffic adaptive algorithm based on an online RL technique	Direct transition between sleep modes to avoids unnecessary time wastage at intermediate levels	Not consider uplink traffic
	Proposed MSC-PUT	Provide PPO-based Multi-level switching operation	Enhance energy efficiency employing deep sleep mode and consider densely deployment of BSs	-

**Table 2 sensors-23-09070-t002:** Key notations.

Notation	Definition	Notation	Definition
$\Lambda_{m}^{t}$	Total arrival rate for the BS *m* during *t*	$\lambda_{k}^{t}$	Arrival rate of the IoT device *k* during *t*
$\Delta t_{m}$	Light sleep time of the BS *m*	$t_{syn}$	Synchronization time between IoT devices and SBSs
$D_{km}^{t}$	Average latency of the IoT device *k* that connected the BS *m* at time time slot *t*	$\alpha_{m}^{t}$	Operational status of BS *m* at time slot *t*
$D_{km}^{t,LS}$	Light sleep latency of the IoT device *k* that connected the BS *m* at time slot *t*	$D_{km}^{t,HO}$	Handover latency of the IoT device *k* that connected the BS *m* at time slot *t*
$D_{km}^{t,TX}$	Transmission latency of the IoT device *k* that connected the BS *m* at time slot *t*	$N_{k}^{data}$	Amount of uplink transmission from the IoT device *k*
$\rho_{m}^{t}$	Traffic load of BS *m* at time slot *t*	$\gamma_{km}^{t}$	SINR value between IoT device *k* and the serving BS *m* at time slot *t*
$N_{k}^{sense}$	Sensed data per second of the IoT device *k*	$c_{k}$	Transmission periodicity of the IoT device *k*
$t_{arr}$	Arrival time of uplink data	$t_{de}$	Ending time of deactivation state from active state
$t_{ls}$	Ending time of light sleep state	$t_{re}$	Ending time of reactivation state from light sleep state
$d_{de}$	Latency of arrived data at deactivation state	$d_{ls}$	Latency of arrived data at light sleep state
$d_{re}$	Latency of arrived data at reactivation state	*r*	Residual time for active state
$\Delta t_{de}$	Deactivation duration	$\Delta t_{ls}$	Light sleep duration
$\Delta t_{re}$	Reactivation duration	$D^{LS}$	Average light sleep latency
$R_{km}^{t}$	Throughput of uplink data from the IoT device *k* to the BS *m* at time slot *t*	$E_{total}^{T}$	Total energy consumption of BSs during time *T*
$E_{MBS}^{T}$	Energy consumption of the MBS during time *T*	$E_{SBS,m}^{T}$	Energy consumption of the SBS *m* during time *T*
$P_{MBS}^{static}$	Static operational power of the MBS	$P_{MBS}^{dyn\_max}$	Maximum dynamic operational power of the MBS
$\rho_{MBS}^{max}$	Maximum traffic load that MBS can serve	Δt	Slot time
$p_{m}^{t}\left(A\right)$	Active probability of the SBS *m* at time *t*	$P_{SBS}^{active}$	Power of the SBS in the active state
$P_{SBS}^{ls}$	Power of the SBS in the light sleep state	$P_{SBS}^{ds}$	Power of the SBS in the deep sleep state
$P_{SBS}^{static}$	Static operational power of the SBS	$P_{SBS}^{dyn\_max}$	Maximum dynamic operation power of the SBS
$\rho_{SBS}^{max}$	Maximum traffic load that an SBS can serve	μ	Average service rate of an SBS

**Table 3 sensors-23-09070-t003:** Sleep mode duration [13,16].

Sleep Mode	SM1	SM2	SM3	SM4
Activation/deactivation duration	35.5 μs	0.5 ms	5 ms	0.5 s
Minimum sleep duration	71 μs	1 ms	10 ms	1 s
Corresponding component	OFDM symbol	sub-frame	frame	long-term sleep

**Table 4 sensors-23-09070-t004:** Simulation parameters.

Parameters	Value
Carrier frequency, $f_{c}$	3.5 GHz
Channel bandwidth	40 MHz
BS height, $h_{B}$	8 m
IoT device height, $h_{U}$	1.5 m
Hall size	350 m × 150 m
Number of BSs	1 MBS and 18 SBSs
Number of IoT devices	{10,20,30,40,50}
Data collecting rate	1 Mbits/s
Average number of arrivals	{1,10,50,100,500,1000}
MBS power in active mode	static: 114.5, dynamic: 558.1 W
SBS power in active mode	static: 13.2, dynamic: 7.5 W
SBS power in sleep mode	light sleep: 8.22, deep sleep: 3 W
Transmit power of IoT devices	23 dB
IoT devices’ mobility speed	1 m/s
Handover latency	100 ms
Path loss, PL ($d_{3D}$ in *m*)	$32.4+23\log_{10}\left(d_{3D}\right)+20\log_{10}\left(f_{c}\right)$ dB
Path loss, $PL_{LOS}$ ($d_{3D}$ in *m*)	$31.84+21.5\log_{10}\left(d_{3D}\right)+19\log_{10}\left(f_{c}\right)$ dB
Path loss, $PL_{NLOS}$	$PL_{NLOS}=max\left(PL,PL_{LOS}\right)$ dB
Thermal Noise	−174 dBm/Hz
Number of time slots, *T*	200
Sub-slot time	{5,10,20,40,80,160} ms
Minimum throughput requirement,	1 Mbps
Discount factor, γ	0.99
Learning rate,	$10^{-5}$
GAE lambda,	0.95
Policy clip,	0.2
Batch size	64

**Table 5 sensors-23-09070-t005:** Performance comparison with proposed MSC-PUT.

	MSC-PUT	AlwayOn	Light	NoConn	Threshold	Binary
Throughput	41.97	44.10	43.77	43.77	43.38	33.53
	100.00%	95.16%	95.87%	95.87%	96.75%	125.16%
Energy Consumption	231.61	408.21	307.94	302.68	259.35	208.54
	100.00%	56.74%	75.21%	76.52%	89.30%	111.06%
Energy Efficiency	0.181	0.108	0.142	0.145	0.167	0.161
	100.00%	168%	127.51%	125.31%	108.28%	112.74%

## Data Availability

Data are contained within the article.

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
