# Peer review of "An Energy-Efficient Multi-Level Sleep Strategy for Periodic Uplink Transmission in Industrial Private 5G Networks"

_sensors, 2023, doi:10.3390/s23229070_

Round 1

Reviewer 1 Report

Comments and Suggestions for Authors

I have reviewed the manuscript carefully entitle “An Energy-Efficient Multi-Level Sleep Strategy for Periodic Uplink Transmission in Industrial Private 5G Networks”. The topic is useful and interesting but manuscript needs some update before acceptance. My concerns are given below.

·       Abstract is too general and needs revision with pacific more numeric information.

·       Table of abbreviation table of Key notation should be moved after abstract before start of text.

·       Authors claimed 40 percent improvement related to conventional BS ON-OFF scheme.

It is huge break through if so, I think the authors should re-evaluate the values or statement. On what ground this claim is, it need to describe in the form of table with all parameters.

·       Manuscript is lack of focus discussion and authors need to add discussion section before conclusion.

·       Table 1 should move as Bench mark table in discussion section by giving compression of current study with literature by giving its pros and cons. Authors should add more parameters I the table that show the accuracy of model.

·       Conclusion is also generic and need to add concluding remarks with significance

Comments on the Quality of English Language

need to check

Reviewer 2 Report

Comments and Suggestions for Authors

In the article, “An Energy-Efficient Multi-Level Sleep Strategy for Periodic Uplink Transmission in Industrial Private 5G Networks”. So the author must give a good idea of the importance of using the proposed Efforts to enhance the EE of BSs within ultra-dense networks that have predominantly centered on the development of sleep mode control algorithms. However, I have major comments and suggestions for the authors as follows:

1-               The authors must give the abbreviation for "5G" in the abstract, and “us” in table 3.

2-               In section 3.1.1 Average latency, the authors didn't support the citation reference for equations.

3-               In section 3.1.1 Average latency, the authors didn't support the citation reference for equations.

4-               In algorithm 1: the loop for "for" needs to be revised and written correctly.

5-               The authors don't provide any contribution in section 4. Proposed Method, compare with different latest studies.

6-               The authors must make a comparison between Sleep mode Control for Periodic Uplink Transmission (MSC-PUT) and the previous algorithm to check the enhancement.

7-               The authors must support the convenience answer to enhance energy saving in section results based on applying "the novelty for energy-efficient Multi-level Sleep mode Control for Periodic Uplink Transmission (MSC-PUT).

8-               The authors in the section conclusions explain Reinforcement learning without any explanation of the system model.

9-               The article lacks novelty. Therefore, it must be clarified what is the main part that has been developed as shown in the abstract.

Comments on the Quality of English Language

In the article, “An Energy-Efficient Multi-Level Sleep Strategy for Periodic Uplink Transmission in Industrial Private 5G Networks”. So the author must give a good idea of the importance of using the proposed Efforts to enhance the EE of BSs within ultra-dense networks that have predominantly centered on the development of sleep mode control algorithms. However, I have major comments and suggestions for the authors as follows:

1-               The authors must give the abbreviation for "5G" in the abstract, and “us” in table 3.

2-               In section 3.1.1 Average latency, the authors didn't support the citation reference for equations.

3-               In section 3.1.1 Average latency, the authors didn't support the citation reference for equations.

4-               In algorithm 1: the loop for "for" needs to be revised and written correctly.

5-               The authors don't provide any contribution in section 4. Proposed Method, compare with different latest studies.

6-               The authors must make a comparison between Sleep mode Control for Periodic Uplink Transmission (MSC-PUT) and the previous algorithm to check the enhancement.

7-               The authors must support the convenience answer to enhance energy saving in section results based on applying "the novelty for energy-efficient Multi-level Sleep mode Control for Periodic Uplink Transmission (MSC-PUT).

8-               The authors in the section conclusions explain Reinforcement learning without any explanation of the system model.

9-               The article lacks novelty. Therefore, it must be clarified what is the main part that has been developed as shown in the abstract.

Reviewer 3 Report

Comments and Suggestions for Authors

This article provides and energy efficient architecture for deploying multiple levels of sleep modes for base stations, focusing on 5G deployment in Industrial IoT scenarios. The work is presented well, with the authors providing sound overview of the state of the art, clearly defining the contributions of this article. The results presented show that the reinforcment learning-based approach improves the overall energy efficiency of the base stations, with minor effects on throughput. Minor aspects of this article that could be improved are related to the use of many abreviations, which might create a difficulty to follow the article. In addition, Fig. 4 could be improved by increasing the marker size and improving the positioning of the text labels. Regarding the discussion/conclusion, it would be of interest to provide a brief reflection on how this throughput deterioration would affect actual mixed-criticality industrial environments.

Reviewer 4 Report

Comments and Suggestions for Authors

An Energy-Efficient Multi-Level Sleep Strategy for Periodic Uplink Transmission in Industrial Private 5G Networks

The paper addresses an important and timely problem of energy efficiency (EE) in industrial private 5G networks, where base stations (BSs) and internet of things (IoT) devices are densely deployed. The paper proposes a novel multi-level sleep strategy for periodic uplink transmission (MSC-PUT) that leverages reinforcement learning (RL) to optimize EE and network performance.

Overall, this is a well-written and high-quality paper that makes a significant contribution to the field of EE in industrial private 5G networks. The paper is well-organized, well-structured, and well-referenced. The paper is also clear, concise, and coherent in its presentation and argumentation. The paper demonstrates a thorough understanding of the problem and its challenges, as well as a creative and effective solution using RL. The paper also provides rigorous mathematical modeling and empirical evaluation of the proposed solution.

Some minor suggestions for improvement are:

1. The paper could provide more details on how the PPO algorithm is trained and tuned, such as the hyperparameters, reward function, policy network architecture, etc.

2. The paper could comment and examine the PPO algorithm complexity as it involve a learning process.

3. The paper could discuss some potential limitations or drawbacks of the proposed solution, such as scalability, robustness, or security issues, as well as some possible future work or extensions.

Round 2

Reviewer 1 Report

Comments and Suggestions for Authors

The revision performed by the authors is satisfactory, therefore i endorsed the manuscript for publication

Comments on the Quality of English Language

ok

Author Response

We would like to express our sincerest gratitude for the time and effort you have dedicated to the evaluation of our manuscript, which has resulted in significant improvements to its quality.
We sincerely appreciate your invaluable feedback and constructive suggestions, which have undoubtedly contributed to the enhanced version of our paper. Thank you once again for your dedication and valuable support throughout this review process.

Reviewer 2 Report

Comments and Suggestions for Authors

Recommendations

In the article, “An Energy-Efficient Multi-Level Sleep Strategy for Periodic Uplink Transmission in Industrial Private 5G Networks”. So the author must revise and update the comments as follows:

1- The authors must give the full word for the "5G" abbreviation in the abstract.

2- Eq.( 28) is incomplete in terms of rewards and needs improvement.

Comments on the Quality of English Language

Recommendations

In the article, “An Energy-Efficient Multi-Level Sleep Strategy for Periodic Uplink Transmission in Industrial Private 5G Networks”. So the author must revise and update the comments as follows:

1- The authors must give the full word for the "5G" abbreviation in the abstract.

2- Eq.( 28) is incomplete in terms of rewards and needs improvement.
